# Reference Values of Cerebral Artery Diameters of the Anterior Circulation by Digital Subtraction Angiography: A Retrospective Study

**DOI:** 10.3390/diagnostics12102471

**Published:** 2022-10-12

**Authors:** Dirk Halama, Helena Merkel, Robert Werdehausen, Khaled Gaber, Stefan Schob, Ulf Quäschling, Svitlana Ziganshyna, Karl-Titus Hoffmann, Dirk Lindner, Cindy Richter

**Affiliations:** 1Department of Oral and Maxillofacial Surgery, University of Leipzig Medical Center, 04103 Leipzig, Germany; 2Department of Neuroradiology, University of Leipzig Medical Center, 04103 Leipzig, Germany; 3Department of Anesthesiology and Intensive Care Medicine, University of Leipzig Medical Center, 04103 Leipzig, Germany; 4Department of Neurosurgery, University of Leipzig Medical Center, 04103 Leipzig, Germany; 5Department of Radiology, Halle University Hospital, 06120 Halle, Germany; 6Department of Radiology, Kantonsspital Baselland, 4410 Liestal, Switzerland; 7Transplant Coordinator Unit, University of Leipzig Medical Center, 04103 Leipzig, Germany

**Keywords:** vessel diameter, reference value, effect size, nomogram, morphological variation, digital subtraction angiography

## Abstract

A threshold-based classification of cerebral vasospasm needs reference values for intracranial vessel diameters on digital subtraction angiography (DSA). We aimed to generate adjusted reference values for this purpose by retrospectively analyzing angiograms and potential influencing factors on vessel diameters. Angiograms of the anterior circulation were evaluated in 278 patients aged 18–81 years. The vessel diameters of 453 angiograms (175 bilateral) were gathered from nine defined measuring sites. The effect sizes of physical characteristics (i.e., body weight and height, body mass index, gender, age, and cranial side) and anatomical variations were calculated with MANOVA. Segments bearing aneurysms were excluded for the calculation of reference values. Adjusted vessel diameters were calculated via linear regression analysis of the vessel diameter data. Vessel diameters increased with age and body height. Male and right-sided vessels were larger in diameter. Of the anatomical variations, only the hypoplastic/aplastic A1 segment had a significant influence (*p* < 0.05) on values of the anterior cerebral artery and the internal carotid artery with a small effect size (|ω^2^| > 0.01) being excluded from the reference values. We provide gender-, age-, and side-adjusted reference values and nomograms of arterial vessel diameters in the anterior circulation.

## 1. Introduction

Digital subtraction angiography (DSA) is considered the gold standard imaging modality in acute cerebrovascular diagnosis. Its role has become increasingly prominent, since the endovascular therapy of acute ischemic stroke is a standard of care procedure. It provides diagnostic information regarding intracranial arterial diseases, e.g., atherosclerosis, stenosis, vessel dissection, vasculitis, moyamoya disease and vascular malformations [1]. The requirement of ionizing radiation and the procedural risks of an invasive procedure make it suboptimal for routine vascular assessment. Therefore, data acquisition from healthy subjects in order to create a reference group is challenging.

DSA is superior to computed tomography angiography (CTA) in assessing the severity and the impact of symptomatic cerebral vasospasm (CVS) on subsequent perfusion [2] with indicated endovascular intervention [3]. Mild and moderate vasospasms in distal locations are detected less on CTAs. Furthermore, CTA can overestimate the degree of CVS by underestimating the diameter of large cerebral arteries. In previous studies [4,5], the diagnosis was usually made by comparing the DSA on admission with the DSA at the time of suspected CVS. Early angiographic vasospasm on admission was not taken into account [6]. The most commonly used scheme to classify CVS is a reduction in vessel diameters of <25% (mild), 25–50% (moderate), and >50% (severe) [7,8]—or <30% (grade 1), 30–70% (grade 2), and >70% (grade 3) [9]. Reference values for intracranial arteries are required to standardize CVS classification [10]. There are only a few longitudinal retrospective studies regarding cerebral arterial morphology in DSA but without detailed measurements of vessel diameters [11], which is likely related to the difficulty in assessing the influence of local morphological particularities on vessel diameters. The most common morphological changes of cerebral vessels are encountered in their origins, calibers, communications and branching. We assessed the influence of the most common anatomical variations (i.e., fetal variation, A1 aplasia/hypoplasia, early middle cerebral artery (MCA) branching, and two versus three MCA branches) and aneurysm locations (i.e., anterior communicating artery (AComA) aneurysms, internal carotid artery (ICA) aneurysms, and MCA branching aneurysms) on intracranial vessel diameters at nine defined measuring sites and identified size influencing factors. After excluding falsifying influencing factors, we generated nomograms for intracranial vessels to allow metrically standardized vasospasm classification.

## 2. Materials and Methods

### 2.1. Study Design and Recruitment

This retrospective study was performed according to the “Strengthening the Reporting of Observational Studies in Epidemiology (STROBE)” guidelines [12]. Key word research (e.g., DSA and ICA) in our radiology information system (RIS) revealed three thousand six hundred and eighteen diagnostic and interventional cerebral digital subtraction angiographies (DSAs) performed in the Department of Neuroradiology at the University of Leipzig Medical Center between October 2007 and September 2019. Only diagnostic DSAs with extracranial catheter locations were included in our study to gather reference values. A detailed flow chart with the numbers of the excluded DSAs is shown in Appendix A.

Inclusion criteria for the study population:Age equal or older than 18 years;Diagnostic DSA of the internal carotid artery;Control DSA in cases of SAH after 6 months or later after SAH.

Exclusion criteria for the study population:Age under 18 years;Any neurological event in the last six months to DSA;Oral, intravenous or intraarterial nimodipine administration;Moderate or severe arteriosclerosis on CT;Mycotic- or traumatic-induced pseudoaneurysms and aneurysms associated with arteriovenous malformations;Stenosis or dissections of intra- or extracranial arteries belonging to the cerebral circulation;Diseases affecting the intracranial vessels, e.g., moyamoya disease, fibromuscular dysplasia, and basilar artery megadolychoectasia;Territorial infarction or intracerebral hemorrhage after SAH;Any malignant tumor;Any intracranial space-occupying lesion with impact on vessel diameters.

### 2.2. Diagnostic Transcranial Angiography

Diagnostic DSA was performed using a biplane system (Axiom Artis, Siemens, Erlangen, Germany and AlluraClarity, Philips Healthcare, Best, The Netherlands) or a monoplane system (Innova 4100; GE Healthcare, Waukesha, WI, USA). Iopromid (60–120 mL, containing 300 mg iodine per mL) was used as the contrast agent.

DSA was performed according to vascular regions with suspected CVS. Therefore, not all segments were necessarily examined in every patient, especially not the vertebrobasilar circulation.

### 2.3. Anatomical Variations

All angiograms were evaluated for anatomical variations before angiographic measurements by one consulting physician in neuroanatomy and radiology with 3 years of angiographic experience. Ambiguous angiograms were discussed and double checked by an interventional neuroradiologist with more than three years of experience. The proximal segments of the posterior communicating artery (P1) or the anterior cerebral artery (A1) with diameters smaller than 0.5 mm were considered “hypoplastic”, and the absence of the artery was “aplastic”. These congenital variations of the circle of Willis were the most common ones. The P1 hypoplasia or aplasia with a prominent posterior communicating artery (PComA) is known as the “fetal-type” origin of the PCA. An aplastic or hypoplastic A1 segment is often named the anterior variation. Hypoplastic or aplastic variations of the same segment were pooled for statistical evaluation. An early MCA bifurcation was defined by a prominent M2 branch originating not in the last third of the straight portion of the M1 segment.

### 2.4. Angiographically Measured Sites

We only focused on the anterior circulation by evaluating internal carotid artery angiograms. In the lateral projection, the arterial diameters of the extradural internal carotid artery (ICA: C5) and intradural internal carotid artery (ICA: C6) were measured. The remaining segments of the intradural internal carotid artery (ICA: C7), the middle cerebral artery (MCA: proximal M1 = pM1, distal M1 = dM1, and M2), and the anterior cerebral artery (ACA: proximal A1 = pA1, distal A1 = dA1, and A2) were measured in the posterior–anterior projections (Figure 1).

All values were only measured on a diagnostic workstation (syngo.plaza, VB30C, Siemens Health Care, Erlangen, Germany) according to a defined protocol:

In the lateral projection:-C5: 2 mm proximal to the origin of the ophthalmic artery;-C6: At the origin of the ophthalmic artery.

In the posterior–anterior projection:-C7: 2 mm proximal to the carotid T;-pM1: 2 mm distal to carotid T;-dM1: 2 mm proximal to middle cerebral artery bifurcation;-M2: 4 mm distal to the M1/M2 transition;-pA1: 2 mm distal to carotid T;-dA1: 2 mm proximal to AComA origin or change of course to A2;-A2: 2 mm distal to AComA origin or change of course from A1.

The investigator assessed whether vessel narrowing was a preexisting condition, such as developmental hypoplasia or atherosclerosis. The proximal ACA was considered hypoplastic if the contralateral A1 segment and the AComA were large and feeding both A2 segments. In the case of the proximal bifurcation variant of the MCA, the most prominent branch of the M1 segment was evaluated. The two most prominent M2 branches were measured if there was a trifurcation or quattrofurcation. The values were not measured at the site of aneurysms or implanted materials such as coils.

Exclusion criteria for vessel diameters:Atherosclerotic changes;Aplastic or hypoplastic A1 segments (exclusion of C5, C6, C7, pA1, dA1, and A2 segment values for both sides);ACA trifurcation (A2 segment values);Duplicate M1 segment (exclusion of pM1 and dM1 values);Segment fenestration (not present);Segments bearing an aneurysm:ICA/PComA/trigeminal artery aneurysms: C5, C6, and C7;ACA/AComA aneurysms: dA1, pA1, and A2;M1/MCA bifurcation: pM1, dM1, and M2.

### 2.5. Physical Characteristics

For statistical analysis of the influencing parameters, we divided the vessel diameters into subgroups according to body height, body weight, body mass index (BMI), age, gender, and cranial side. The mean height of the German population is 180 cm [13], which was chosen as the cut-off value for body height. Total obesity is defined as a BMI ≥ 30 kg/m^2^ [14], serving as a cut-off value. A body weight of 100 kg was calculated for a BMI of 30 kg/m^2^ and body height of 180 cm as the nearest cut-off value. A cut-off value of 50 years was the result of clustering the vessel values according to age. Data clustering is a method that can form classes of objects with similar characteristics. This method partitioned the dataset of vessel diameters into clusters. According to a defined age, BMI, and body height, diameters in the same cluster were more similar than diameters in different clusters.

The following subgroups were defined: body height < 180 cm versus body height ≥ 180 cm, body weight < 100 kg versus body weight ≥ 100 kg, BMI < 30 kg/m^2^ versus BMI > 30 kg/m^2^, age < 50 years versus age ≥ 50, female versus male, and left versus right.

### 2.6. Calculation of Reference Values

Linear regression was calculated with the revised vessel diameters (independent values) to predict the gender-, age-, and side-dependent reference values (fitted values). The statistical variance is quoted as a 95% confidence interval. The lower and upper bounds are named confidence bounds.

### 2.7. Statistical Analysis

Statistical analyses were performed with SPSS version 27.0 (IBM Corporation; New York, NY, USA) and R version 4.0.1 (The R Foundation for Statistical Computing, Indianapolis, IN, USA).

After testing for normal distribution, data were analyzed for the effect size with R using a 2-tailed multivariate analysis of variance (MANOVA). A *p-*value < 0.05 was considered to indicate statistical significance, which was graduated as follows: *p*-value < 0.05 (significant), *p*-value < 0.01 (highly significant), and *p*-value < 0.001 (extremely significant).

The statistical power of a test is the probability of correctly rejecting the null hypothesis [15]. It depends on the sample size of the study, the size of the effect, and the significance criterion [16]. The effect size highlights the importance of the practical significance of our results. First, we wanted to know whether the anatomical variation or aneurysm location had an effect greater than zero and how large was the effect. A major goal of developing effect size measures is to provide a standard metric. Effect sizes can be interpreted by meta-analysts and others across studies that vary in their dependent variables as well as types of designs.

The best-known effect size parameter is Cohen’s partial eta squared (η^2^p), belonging to the d family (consisting of standardized mean differences). We used Cohen’s omega squared (ω^2^) belonging to the r family (measures of strength of association). It is less biased than eta squared [16]. The r family effect sizes describe the proportion of variance that is explained by group membership. A correlation of 0.5 indicates that 25% (ω^2^) of the variance is explained by the difference between groups. The effect size is calculated from the sum of squares for effect divided by the sums of squares for other factors in the design [16]. The difference between individual observations as well as the mean for the group were discounted.

The Cohen’s omega squared (ω^2^) measures the strength of an association between different terms based on the entire dataset |ω^2^|: ≥0.010 = small, ≥0.100 = medium, and ≥0.250 = large. The absolute value of omega squared was used.

## 3. Results

### 3.1. Study Population

Four hundred and fifty-three angiograms (175 bilateral) of 278 patients fulfilled the inclusion criteria. General demographic data are summarized in Table 1. The patients were predominantly female (67.3%). More than half of the patients aged 18–81 (mean: 55.5 years) obtained a bilateral diagnostic DSA (62.9%). The majority (94.6%) suffered from intracranial or extracranial aneurysms. The proportion of incidental aneurysms predominated with 60.1% versus 34.5% of cases with previous SAH (>6 months ago). In one case, no bleeding source was found. Among all cases, 44.2% had more than one aneurysm. In only 15 patients (30 angiograms, 6.6%), DSA revealed no vascular disease.

First, the means of vessel diameters were compared with the ANOVA for patients with and without aneurysms and did not show any significant difference (C5: *p* = 0.138, C6: *p* = 0.275, C7: *p* = 0.391, pM1: *p* = 0.794, dM1: *p* = 0.596, M2: *p* = 0.389, pA1: *p* = 0.396, dA1: *p* = 0.698, and A2: *p* = 0.715). Furthermore, vessel diameters of patients, who suffered from a SAH (more than six months ago), did not significantly differ from those with incidental aneurysms (C5: *p* = 0.662, C6: *p* = 0.614, C7: *p* = 0.913, pM1: *p* = 0.978, dM1: *p* = 0.862, M2: *p* = 0.714, pA1: *p* = 0.232, dA1: *p* = 0.179, and A2: *p* = 0.227).

### 3.2. Anatomical Variations

The anatomical variations of all angiograms are summarized in Table 1. Congenital abnormalities were demonstrated in 26.5% (n = 120) of angiograms. Anterior cerebral artery variations with aplastic or hypoplastic A1 segments were most frequently found with 14.1% (left: n = 28, 6.2%; right: n = 34, 7.5%). Variations of the PComA with hypoplastic or aplastic P1 segments (fetal type) followed with 12.4% (left: n = 24, 5.3%; right: n = 32, 7.1%). Two right-sided angiograms (0.4%) had a persistent trigeminal artery as the most common congenital carotid-vertebrobasilar anastomosis. The most common variation on vessel branching was an early MCA bifurcation in 42.8% (n = 194) with a slight predominance for the left side (left: n = 105, 23.2%; right: n = 89, 19.6%) of angiograms. A doubled M1 segment was a rare finding in only two left-sided angiograms (0.4%) as well as an ACA trifurcation in one right-sided angiogram (0.2%).

### 3.3. Influence of the Physical Characteristics and Anatomical Variations on the Vessel Diameters

We evaluated the statistical influence of physical characteristics (Table 2) and anatomical variations (Appendix A) on vessel values for each measured point with a MANOVA-like effect size calculating the *p*-values and |ω^2^|-values. The effect size represents the amount of variance explained by each model’s terms (i.e., body height, body weight, BMI, gender, size, age, anatomical variation, and number of MCA branches), where each term can be represented by one or more parameters. Only factors with significant *p*-values < 0.05 and at least a small effect size of|ω^2^| > 0.01 were estimated as real influencing factors. The vessel diameters of tiny segments are hard to measure with a wide range of values. *p*-Values strongly depend on the sample size; therefore, the effect size was chosen as a second criterion for validation. Highly significant *p*-values without any measurable effect size were rated insignificant.

Body height, body weight, BMI, age, gender, and cranial side were evaluated for their statistical influence on vessel values (Table 2). Especially, the large segments of the carotid T (e.g., C7, pM1, and pA1) were dependent on body height, whereas body weight and BMI had no relevant impact on vessel diameters. The diameters of the ICA and MCA segments strongly depended on age. The ACA segments were influenced by cranial side. ACA and MCA segments showed significant differences with smaller values for females.

Anatomical variations (i.e., A1 hypoplasia/aplasia, fetal-type variation, early M2 branching, and two versus three MCA branches) in ten or more angiograms were evaluated for their statistical influence on vessel values (Appendix A). Vessel values with a significant *p*-value (<0.05) and |ω^2^| > 0.01 (small prediction) or *p* < 0.01 (highly significant) or |ω^2^| > 0.06 (intermediate prediction) for the anatomical variation were excluded from further analysis (Appendix A).

The cranial side had an impact on MCA branching variation and the anterior cerebral artery variation with A1 hypoplasia or aplasia and the fetal-type variation.

The anterior cerebral artery variation with A1 hypoplasia or aplasia was excluded, since it revealed high significance and a low prediction for C5, C6, pA1, dA1, and A2 segments. The C7 segment, as part of the ICA, was excluded as well. Testing for the M2 branching and fetal-type variation resulted in a *p*-value of 0.048 in the C7 segment without any prediction.

### 3.4. Adjusted Reference Values and Nomograms of the Intracranial Arteries

The best-fitted line was calculated with the linear regression of vessel diameters for each segment adjusted for age, gender, and cranial side. Figure 2 illustrates the best-fitted lines (blue) of the pM1 segment for gender and cranial side-dependent distribution of vessel values (black points). The grey area indicates the statistical variance with a 95% confidence interval for the best-fitted values of another study group. The lower and upper bounds of the 95% confidence intervals are shown as red lines. In Table 3, the best-fitted values (blue line in Figure 2), the 95% confidence interval for the best-fitted values (red lines in Figure 2), and the single measuring values are listed. One measured value would lie in between these confidence bounds with a 95% probability. The figures of the best-fitted lines of the remaining measuring sites are displayed in Appendix A.

Side- and gender-adjusted best-fitted lines are shown as nomograms for each measuring site in Figure 3.

## 4. Discussion

This study aimed to gather rectified reference values of intracranial artery diameters to facilitate device implantation and vasospasm classification. Thus, we evaluated the influence of the most common anatomical variations, BMI, body height, gender, cranial side, and patient’s age on intracranial vessel diameters at nine defined measuring sites.

### 4.1. Age

The results revealed that increasing age was associated with increasing vessel diameters in nonatherosclerotic vessels. Postmortem studies in humans have indicated that aging leads to increased elastin and collagen accumulation in the vessel wall, along with fibrosis, intimal, and medial thickening [17]. In the pathological process of atherosclerosis, calcified lipid or fatty deposits accumulate circumferentially along with the intimal layer of the vessel wall [18]. As the plaques form, the walls become thicker, fibrotic, and calcified. With the narrowing of the lumen, the blood flow increases. These changes and the accumulation of atherosclerotic plaque decrease vessel distensibility. Aging is a major risk factor for cardiovascular disease. For anatomical reasons, intracranial vessels on the right body are more exposed to pulse pressure. A large clinical trial showed that mean arterial pressure increases with age until around 70 years for both sexes [19]. The extracranial arteries are distensible with little change in the stiffness of their walls under 30 or 40 years of age [20]. On the contrary, the intracranial arteries are already stiffer at birth and gradually become even stiffer over a lifetime, which results in a large difference between intra- and extracranial arteries in middle age [20].

Histochemical and electron microscopic studies have demonstrated a cholinergic perivascular nerve plexus in the wall of cerebral arteries [21]. Saba et al. [22] showed a significant reduction of 30–45% in the density of the noradrenergic and acetylcholinesterase-positive innervation of the MCA with old age. The sympathetic innervation of cerebral arteries is involved in the autoregulation of cerebral blood flow (CBF). The sympathetic nerves attenuate any increases in CBF once the sympathetic regulatory capacity is surpassed, especially in severe hypertension [23].

### 4.2. Influence of Gender, Cranial Side, Body Mass Index, and Height

In our study, left-sided and male vessels were larger in diameter than female and right-sided vessels. Males are usually taller than females, thereby influencing the primary vessel constitution. Cranial side [24,25,26,27,28,29,30,31,32] and gender [32,33] differences for intracranial arteries are already described in the literature. The variant vessel diameters in former studies came from undetermined measuring sites and different methods. There is a controversy in the literature, as Muller et al. [32] found no correlation of vessel diameters with age in DSA, except for the right MCA (n = 261) with an increase of 7.8% from the fifth to the sixth decennium in males. The ACA remained unchanged. As the most vulnerable artery to atherosclerosis of the intracranial arteries, the ICA was not evaluated. In a study by Rai et al. [29], the most considerable variation was seen in the length of the MCA. There was no significant difference between the values based on either gender or cranial side in accordance with our study for MCA aneurysms. Additionally, the diameter at the M1 origin was significantly larger in older patients. Patients aged >60 years had significantly longer ICA segments between the proximal cavernous carotid artery and the ICA terminus.

Reference values for the aortic diameter have been reported in several studies and countries [34]. Hu et al. [35] investigated the influence of age, gender, BMI, and body surface area (BSA) on the diameter of the abdominal aorta and the common iliac arteries of middle-aged and elderly people in China (n = 625). The vessel diameters were significantly higher in males than females and increased with age. The diameters were positively correlated with BSA, body height, body weight, and BMI. Selim et al. [36] determined the effects of BMI on cerebral blood flow regulation in patients with type 2 diabetes mellitus (n = 30), hypertension (n = 45), and stroke (32) versus healthy controls (n = 90) in transcranial Doppler. The MCA and ICA diameters for both sides were not different among the four groups. Men, especially those with stroke, had a lower mean blood flow velocity than women. High BMI was associated with reduced cerebral blood flow velocity and increased cerebrovascular resistance. To date, the influence of physical characteristics on cerebral vessel diameter has not been investigated.

Women have a higher lifetime risk for stroke because of their longer life expectancy and much higher incidence of stroke at an older age. They have more events and are less likely to recover from stroke events [37], although at a lower age, stroke is more common among men. Women are smoking less commonly than men, and the protective effect of estrogen on cerebral vasculature might be another reason. However, hypertension and dyslipidemia were more common in women [38].

Obesity leads to increased arterial stiffening and an increased diameter of the internal carotid artery already in middle childhood or early adolescence [14,39]. Long-term exposure to hemodynamic stimuli and metabolic disturbance caused by obesity augments the arterial impedance and afterload of the heart. As a marker of structural change, we demonstrated that the vessel diameter did not significantly increase in the obese cohort of our study. This might be due to the fact that atherosclerotic changes in intracranial arteries were an exclusion criterion of this study. Moreover, vessel diameters were dependent on body height and gender, supporting rectified reference values for cerebral arteries.

### 4.3. Anatomical Variations

In the present study, the overall rate of congenital variations in the circle of Willis was 26.5%. Concerning anatomical variations, previous studies showed ambiguous findings. Orakdogen et al. [11] analyzed 128 aneurysm cases on DSA angiograms for vascular variations. In accordance with their results (34.4%), in our study group, the anterior cerebral artery variation was the most common finding in aneurysm cases, followed by fetal-type variation (24.2%). Krzyzewski et al. [40] found an incidence of 36% for the hypoplastic and aplastic A1 segment in CTA, whereas Kovac et al. [41] reported an incidence of 17.6% (hypoplastic) and 0.4% (aplastic) in CTA, retrospectively. A hypoplastic/aplastic P1 segment with a prominent PComA varied from 10% [42] in cadaveric studies to 28% [11] on DSA angiograms. A microsurgical study by Gibo et al. [43] revealed that anomalies of the MCA are less common than other anomalies consisting of duplicate M1, confirming our findings. They observed MCA trifurcation in 12% of cases. The different proportions result from different analyzed inclusion and exclusion criteria of the studies. Study populations with patients suffering from intracranial aneurysms have a higher proportion of anatomical variations. Some specific aneurysms can be encountered together with anatomical variations. The hypoplasia of the A1 segment was frequently associated with AComA aneurysms [11], according to our results. In contrast, PComA origin aneurysms did not prevail in fetal-type variations. Macchi et al. proposed for MR angiography that anatomical variation rates were higher on the left side, being controversial to the results of Orakdogen et al. [11] and the results presented here. Van Overbeeke et al. [44] noted that variations occur as developmental modifications more than genetic factors without any influence on the cranial side or gender.

### 4.4. Rectified Reference Values for Standardized CVS Classification

We focused on the anterior circulation diameters because the vertebrobasilar system was infrequently depicted in our cohort. Aneurysms are more commonly located in the anterior (85–90%) than in the posterior circulation (10–15%) [45]. Likewise, vasospasm affects the anterior circulation more often. The reported incidence of vertebrobasilar vasospasm varies from 22.2% [10] to 37.8% [46]. Previous studies investigated the effect of morphological parameters on the risk of aneurysm formation [47]. Vascular anomalies were found angiographically in 88% of patients with multiple intracranial aneurysms [48]. The aneurysms bled proportionally more frequently when associated with a congenital vascular anomaly.

Due to the lack of a standardized CVS classification scheme, previous studies [9,49,50] used variant percentages in vessel diameter decrease, which were calculated by comparing the DSA on admission with the DSA at the time of suspected CVS. None of the threshold-based classification schemes [8,9,50,51] were validated or used adjusted reference values for standardized CVS classification.

The most applied scheme was a reduction of vessel caliber of different intracranial arteries: >50% for severe CVS, between 25–50% for moderate CVS, and <25% for mild CVS [7,8]. Afat et al. [50] and Neulen et al. [50] graded CVS differently according to the following system: 0, no CVS; 1, CVS with <50% change in the vessel diameter; 2, CVS with 50% narrowing compared to the initial DSA. Kerz et al. [9] scored the severity of angiographic CVS with three grades: grade 1 = up to 30%; grade 2 = 30–70%; grade 3 = over 70% reduction of vessel diameter. Weidauer et al. [49] was the first group involving CVS distribution by differentiating between focal (<50% of the segment length) and diffuse CVS (>50% of the segment length). Merkel et al. [10] introduced a DSA-based visual CVS classification with required neuroradiological experience. The threshold-based criterion of their study increased the reproducibility, underlining the need for age-, sex-, and side-adjusted reference values. To date, there is no standardized method to evaluate the effect of intra-arterial pharmacological or mechanical spasmolyses. Our study aimed to provide a basis for a threshold-based classification of cerebral vasospasm. 

#### Limitations

This retrospective study with a focus on imaging findings has several limitations. First of all, a retrospective analysis with key word research in radiology information systems is imprecise. Novel scheduling tools make retrospective data collection easier in future studies. The requirement of ionizing radiation and the procedural risks of an invasive procedure make it challenging to analyze angiograms of healthy humans in order to gain reference values. Most patients suffered from intracranial aneurysms with or without previous SAH. Only a small fraction of 15 patients did not suffer from any vascular disease.

In addition, there may be inter-ethnic variations which are not fully appreciated in this study. Leipzig has an increasing percentage of foreigners and immigrants with 10.8% in 2014, 13.4% in 2016 and 16.8% immigrants in 2021 (State Statistical Office of Saxony) [52]. Most immigrants came from Syria, Ukraine, the Russian Federation, Poland, Iraq, Turkey, Kazakhstan, and Afghanistan. Most of them belong to the European or Asian population. Ethnic minorities carry with them their genetic background and lifestyle habits which merge with high income countries’ cultures, increasing the risk of developing cardiovascular and metabolic diseases [53]. Our study group compromised a small percentage (<5%) of non-European ethnic subgroups with neglectable influence on vessel diameters.

## 5. Conclusions

The present study provides gender-, side-, and age-rectified reference values of intracranial arteries based on nine standardized measuring sites for a European single-center study population. The presented reference values and nomograms provide a basis for a threshold-based classification of CVS with the need for further evaluation.

## Figures and Tables

**Figure 1 diagnostics-12-02471-f001:**
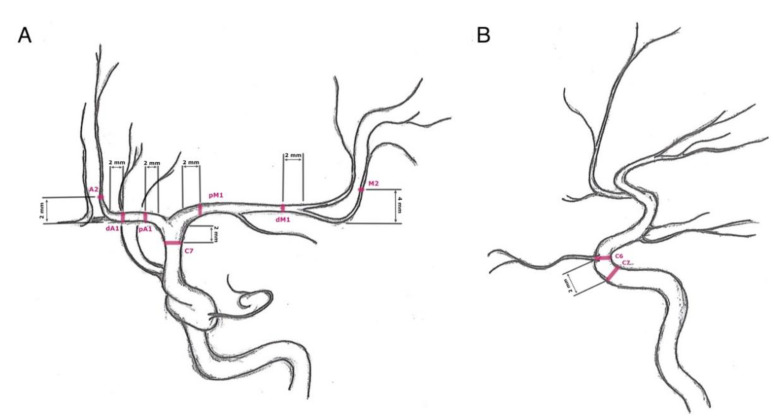
Measuring sites: (**A**) posterior–anterior projection of an internal carotid artery angiogram—C7 (terminal segment of the internal carotid artery): 2 mm proximal to the carotid T, pM1 (proximal horizontal segment of the middle cerebral artery): 2 mm distal to carotid T, dM1 (distal horizontal segment of the middle cerebral artery): 2 mm proximal to middle cerebral artery bifurcation, M2 (insular segment of the middle cerebral artery): 4 mm distal to the M1/M2 transition, pA1 (proximal precommunication segment of the anterior cerebral artery): 2 mm distal to carotid T, dA1 (distal precommunication segment of the anterior cerebral artery): 2 mm proximal to change of course, and A2 (postcommunicating segment of the anterior cerebral artery): 2 mm distal to change of course; (**B**) lateral projection of an internal carotid artery angiogram—C5 (clinoid segment of the internal carotid artery): 2 mm proximal to the origin of the ophthalmic artery and C6 (ophthalmic segment of the internal carotid artery): at the origin of the ophthalmic artery.

**Figure 2 diagnostics-12-02471-f002:**
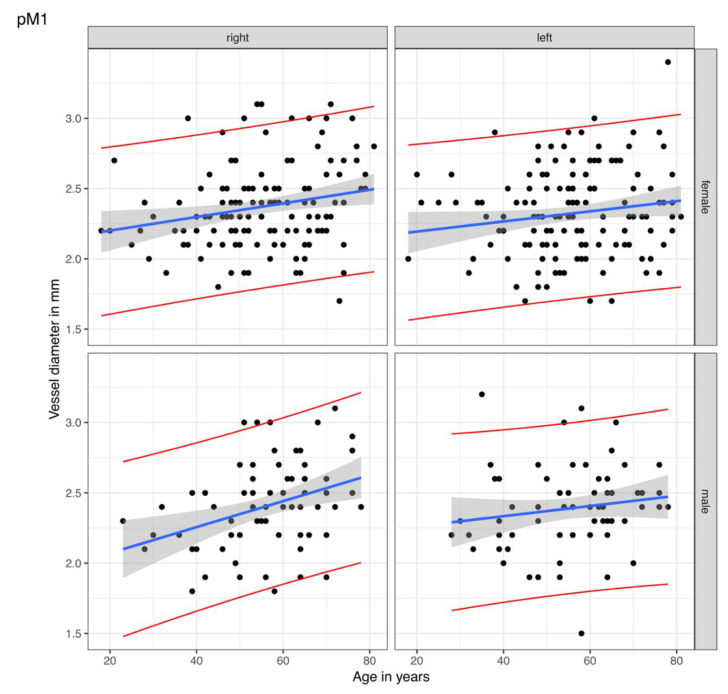
Linear regression of the vessel diameters for the proximal M1 segment. The points are the rectified measured values of the proximal M1 (pM1) segment of the middle cerebral artery in our study group. The blue lines display the calculated best-fitted line with the best-fitted values. The statistical variance is shown by the grey areas as a 95% confidence interval for the best-fitted lines of other study groups, i.e., one single measured value of the corresponding side, gender, and age would lie in between the red lines with a probability of 95%.

**Figure 3 diagnostics-12-02471-f003:**
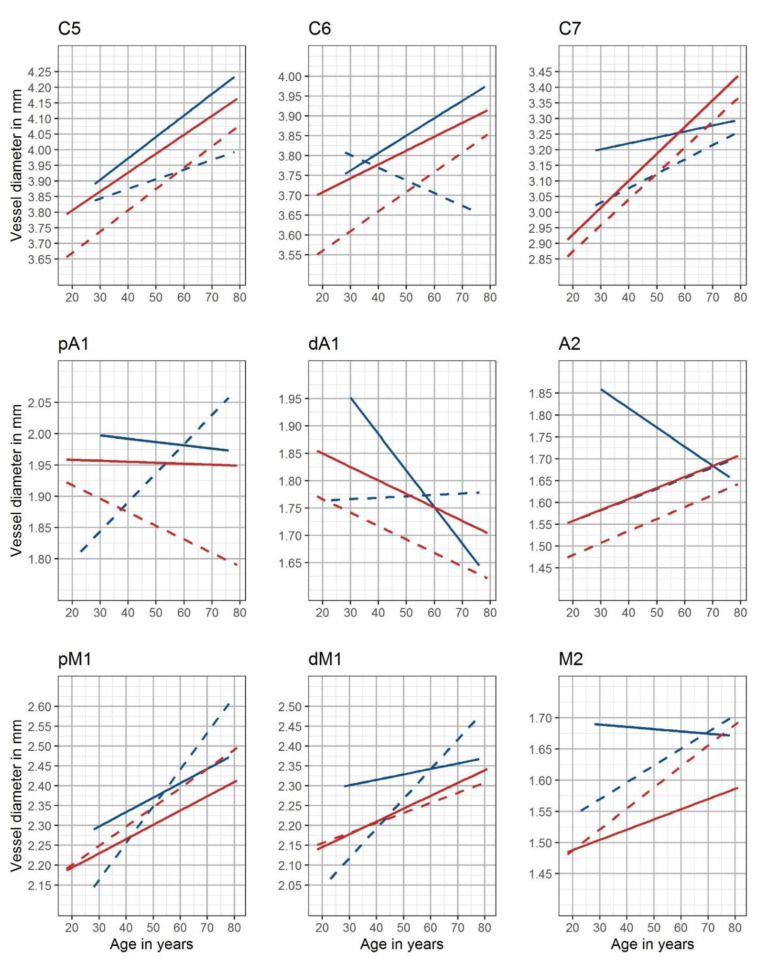
Nomograms of the intracranial arteries. Gender- and side-adjusted nomograms are shown for the nine measuring sites (red: female, blue: male, solid lines: left cranial side, and dotted lines: right cranial side)—C5: clinoid segment of the internal carotid artery; C6: ophthalmic segment of the internal carotid artery; C7: terminal segment of the internal carotid artery; pM1: proximal horizontal segment of the middle cerebral artery; dM1: distal horizontal segment of the middle cerebral artery; M2: insular segment of the middle cerebral artery; pA1: proximal precommunication segment of the anterior cerebral artery; dA1: distal precommunication segment of the anterior cerebral artery; A2: postcommunicating segment of the anterior cerebral artery.

**Table 1 diagnostics-12-02471-t001:** Baseline demographics.

Demographic Data		Location of Aneurysms	
Patients	278	ACA (%)	11 (2.4)
Female (%)	187 (67.3)	AComA (%)	128 (28.3)
Mean age (range)	55.7 (18–81)	ICA (%)	72 (15.9)
**Indication for Diagnostic DSA**		PComA (%)	9 (2.0)
Previous aneurysmatic SAH (%)	96 (34.5)	Trigeminal artery (%)	1 (0.2)
Incidental aneurysm (%)	167 (60.1)	M1 segment (%)	3 (0.7)
Others/no aneurysm (%)	15 (5.4)	MCA bifurcation (%)	39 (8.6)
**Analyzed Angiograms**		VA/BA (%)	73 (16.1)
Total (%)	453 (100.0)	Patients with >1 aneurysm (%)	123 (44.2)
Left (%)	223 (49.2)	**Anatomical Variations**	
Right (%)	230 (50.8)	Fetal type (%)	56 (12.4)
Bilateral (%)	175 (62.9)	Hypoplastic/aplastic A1 (%)	64 (14.1)
**M2 Branches**		Early MCA bifurcation (%)	194 (42.8)
2 (%)	334 (73.7)	Duplicate M1 segment (%)	2 (0.4)
3 (%)	111 (24.5)	ACA trifurcation (%)	1 (0.2)
4 (%)	8 (1.8)	Persisting trigeminal artery (%)	2 (0.4)

**Table 2 diagnostics-12-02471-t002:** Segment specific multivariate analysis of variance (MANOVA) and effect size for vessel diameters (in mm) according to physical characteristics.

		Vessel Diameters in mm								
Group		C5	C6	C7	pM1	dM1	M2	pA1	dA1	A2
**BMI**	*p*-value	0.693	0.881	0.692	0.534	0.322	0.575	0.895	0.810	0.521
	|ω^2^|	0.003	0.003	0.003	0.002	0.000	0.002	0.003	0.003	0.002
BMI < 30 kg/m^2^ (n = 299)	mean	4.0	3.8	3.2	2.4	2.3	1.6	1.9	1.7	1.6
	range	2.5–5.8	2.4–4.9	2.2–4.8	1.7–3.4	1.4–3.2	0.8–2.6	0.7–3.0	0.5–2.6	0.5–2.5
BMI ≥ 30 kg/m^2^ (n = 97)	mean	4.0	3.8	3.2	2.4	2.3	1.6	1.9	1.8	1.6
	range	3.0–5.0	2.9–5.0	2.4–4.8	1.7–3.0	1.6–3.1	1.1–2.3	0.9–3.3	1.1–2.7	1.1–2.2
**Body Height**	*p*-value	0.154	0.110	0.001 **	0.001 **	0.169	0.241	0.033 *	0.105	0.322
	|ω^2^|	0.003	0.005	0.034 *	0.032 *	0.003	0.001	0.011 *	0.005	0.000
<180 cm (n = 339)	mean	4.0	3.8	3.2	2.3	2.2	1.6	1.9	1.7	1.6
	range	2.5–5.8	2.4–5.0	2.2–4.7	1.7-3.4	1.4–3.2	0.8–2.6	0.8–2.9	0.5–2.7	0.7–2.5
≥180 cm (n = 57)	mean	4.1	3.9	3.4	2.5	2.4	1.7	2.0	1.8	1.7
	range	3.0–5.0	3.0–4.8	2.8–4.8	1.9–3.2	1.7–3.2	1.2–2.4	0.7–3.3	0.6–2.5	0.5–2.5
**Body Weight**	*p*-value	0.197	0.383	0.070	0.353	0.758	0.965	0.889	0.714	0.226
	|ω^2^|	0.002	0.001	0.007	0.000	0.003	0.003	0.003	0.003	0.002
<100 kg (n = 358)	mean	4.0	3.8	3.2	2.4	2.3	1.6	1.9	1.8	1.6
	range	2.5–5.8	2.4–5.0	2.2–4.8	1.7–3.4	1.4–3.2	0.8–2.6	0.7–3.0	0.5–2.7	0.5–2.5
≥100 kg (n = 38)	mean	3.9	3.8	3.2	2.4	2.3	1.6	2.0	1.8	1.7
	range	3.0–4.6	3.0–4.7	2.7–4.8	2.0–3.0	2.0–3.1	1.2–2.3	0.9–3.3	1.1–2.3	1.2–2.2
**Age**	*p*-value	0.002 *	0.036 *	0.007 *	0.000 ***	0.022 *	0.042 *	0.969	0.286	0.289
	|ω^2^|	0.027 *	0.011 *	0.020 *	0.038 *	0.013 *	0.010 *	0.003	0.000	0.000
<50 years (n = 138)	mean	3.9	3.7	3.1	2.3	2.2	1.6	1.9	1.8	1.6
	range	3.0–5.0	2.9–4.9	2.3–4.1	1.7–3.2	1.4–3.2	0.8–2.4	1.1–2.8	1.1–2.7	0.7–2.3
≥50 years (n = 315)	mean	4.0	3.8	3.2	2.4	2.3	1.6	1.9	1.7	1.6
	range	2.5–5.8	2.4–5.0	2.2–4.8	1.5–3.4	1.5–3.2	0.9–2.6	0.7–3.3	0.5–2.6	0.5–2.5
**Gender**	*p*-value	0.064	0.101	0.123	0.060	0.005 *	0.001 **	0.044 *	0.037 *	0.024 *
	|ω^2^|	0.008	0.005	0.004	0.008	0.021 *	0.029 *	0.010 *	0.011 *	0.013 *
Female (n = 306)	mean	4.0	3.8	3.2	2.3	2.2	1.6	1.9	1.7	1.6
	range	2.5–5.8	2.4–4.9	2.2–4.7	1.7–3.4	1.4–3.2	0.8–2.6	0.9–2.9	0.8–2.7	0.7–2.5
Male (n = 146)	mean	4.0	3.8	3.2	2.4	2.3	1.6	2.0	1.8	1.6
	range	2.8–5.2	3.0–5.0	2.3–4.8	1.5–3.2	1.7–3.2	0.9–2.4	0.7–3.3	0.5–2.5	0.5–2.5
**Cranial Side**	*p*-value	0.053	0.058	0.116	0.729	0.091	0.492	0.000 ***	0.001 **	0.008 **
	|ω^2^|	0.009	0.008	0.005	0.003	0.006	0.002	0.045*	0.030 *	0.019 *
Left (n = 223)	mean	4.0	3.8	3.2	2.3	2.2	1.6	2.0	1.8	1.7
	range	2.6–5.8	2.6–5.0	2.2–4.8	1.5–3.4	1.5–3.2	0.8–2.6	0.9–3.3	0.8–2.7	0.7–2.5
Right (n = 230)	mean	3.9	3.7	3.1	2.4	2.3	1.6	1.9	1.7	1.6
	range	2.5–5.2	2.4–4.9	2.3–4.7	1.7–3.1	1.4–3.1	0.9–2.4	0.7–2.7	0.5–2.6	0.5–2.3

*p*-Value: <0.05 (significant) *, <0.01 (highly significant) **, and <0.001 (extremely significant) ***; effect size |ω^2^|: ≥0.010 (small) *, ≥0.100 (medium) **, and ≥0.250 (large) ***; BMI, body mass index.

**Table 3 diagnostics-12-02471-t003:** Age-, gender-, and cranial side-adjusted reference values.

Vessel Segment	95% Confidence Interval	Physical Characteristics							
		Female				Male			
		<50 Years		≥50 Years		<50 Years		≥50 Years	
		Left	Right	Left	Right	Left	Right	Left	Right
C5	Fitted value (in mm)	3.9	3.8	4.1	3.9	4.0	3.8	4.1	4.0
	Conf. bounds, fitted values (in mm)	3.8–4.0	3.7–3.9	4.0–4.2	3.9–4.0	3.8–4.1	3.7–4.0	4.0–4.2	3.9–4.1
	Conf. bounds, single value (in mm)	3.0–4.9	2.9–4.7	3.2–5.0	3.0–4.9	3.0–4.9	2.9–4.8	3.2–5.0	3.1–4.9
C6	Fitted value (in mm)	3.8	3.7	3.9	3.7	3.8	3.7	3.9	3.8
	Conf. bounds, fitted values (in mm)	3.7–3.9	3.6–3.8	3.8–4.0	3.7–3.8	3.7–3.9	3.6–3.8	3.8–4.0	3.7–3.9
	Conf. bounds, single value (in mm)	2.9–4.7	2.8–4.6	3.0–4.8	2.9–4.6	2.9–4.7	2.8–4.6	3.0–4.8	2.9–4.7
C7	Fitted value (in mm)	3.1	3.1	3.3	3.2	3.1	3.0	3.3	3.2
	Conf. bounds, fitted values (in mm)	3.0–3.2	2.9–3.2	3.2–3.4	3.1–3.3	3.0–3.2	2.9–3.2	3.2–3.4	3.1–3.3
	Conf. bounds, single value (in mm)	2.3–4.0	2.2–3.9	2.4–4.2	2.3–4.1	2.3–4.0	2.2–3.9	2.4–4.1	2.3–4.1
pM1	Fitted value (in mm)	2.2	2.3	2.4	2.4	2.3	2.3	2.4	2.5
	Conf. bounds, fitted values (in mm)	2.2–2.3	2.2–2.3	2.3–2.4	2.4–2.5	2.2–2.4	2.2–2.4	2.4–2.5	2.4–2.5
	Conf. bounds, single value (in mm)	1.6–2.8	1.7–2.9	1.8–3.0	1.8–3.0	1.7–2.9	1.7–2.9	1.8–3.0	1.9–3.0
dM1	Fitted value (in mm)	2.2	2.2	2.3	2.3	2.3	2.3	2.4	2.3
	Conf. bounds, fitted values (in mm)	2.1–2.3	2.1–2.3	2.2–2.3	2.2–2.3	2.2–2.4	2.2–2.3	2.3–2.4	2.3–2.4
	Conf. bounds, single value (in mm)	1.6–2.8	1.6–2.8	1.7–2.9	1.7–2.9	1.7–2.9	1.7–2.9	1.8–3.0	1.7–2.9
M2	Fitted value (in mm)	1.5	1.6	1.6	1.6	1.6	1.6	1.7	1.7
	Conf. bounds, fitted values (in mm)	1.5–1.6	1.5–1.6	1.5–1.6	1.6–1.6	1.5–1.7	1.6–1.7	1.6–1.7	1.6–1.7
	Conf. bounds, single value (in mm)	1.0–2.1	1.0–2.1	1.0–2.1	1.0–2.1	1.0–2.2	1.1–2.2	1.1–2.2	1.1–2.2
pA1	Fitted value (in mm)	2.0	1.9	1.9	1.8	2.0	2.0	2.0	1.9
	Conf. bounds, fitted values (in mm)	1.9–2.0	1.8–1.9	1.9–2.0	1.8–1.9	1.9–2.1	1.9–2.0	1.9–2.1	1.8–2.0
	Conf. bounds, single value (in mm)	1.4–2.6	1.3–2.5	1.3–2.5	1.3–2.4	1.4–2.6	1.4–2.5	1.4–2.6	1.3–2.5
dA1	Fitted value (in mm)	1.8	1.7	1.7	1.7	1.9	1.8	1.8	1.7
	Conf. bounds, fitted values (in mm)	1.7–1.9	1.7–1.8	1.7–1.8	1.6–1.7	1.8–1.9	1.7–1.9	1.7–1.9	1.7–1.8
	Conf. bounds, single value (in mm)	1.3–2.3	1.2–2.3	1.2–2.3	1.1–2.2	1.3–2.4	1.3–2.3	1.3–2.3	1.2–2.3
A2	Fitted value (in mm)	1.6	1.5	1.7	1.6	1.7	1.6	1.8	1.7
	Conf. bounds, fitted values (in mm)	1.6–1.7	1.5–1.6	1.6–1.7	1.5–1.6	1.6–1.8	1.5–1.7	1.7–1.8	1.6–1.7
	Conf. bounds, single value (in mm)	1.1–2.1	1.0–2.1	1.2–2.2	1.1–2.1	1.2–2.2	1.1–2.1	1.2–2.3	1.2–2.2

The linear regression was calculated with the revised vessel diameters (independent values) to predict the gender-, age-, and cranial side-dependent reference values (fitted values). The lower and upper bounds of the 95% confidence interval for the best-fitted values (bf values) of other study groups are listed as the confidence bounds (conf. bounds), fitted values. A single measured value lies in-between the lower and upper bounds of the 95% confidence interval named conf. bounds, single value.

## Data Availability

The datasets generated and analyzed during the current study are available in the Zenado repository (https://www.zenodo.org (accessed on 23 August 2022) digital object identifier number: 10.5281/zenodo.6750905; further inquiries can be directed to the corresponding author.

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
