# Peer review of "Reference Values of Cerebral Artery Diameters of the Anterior Circulation by Digital Subtraction Angiography: A Retrospective Study"

_diagnostics, 2022, doi:10.3390/diagnostics12102471_

Round 1
Reviewer 1 Report (Previous Reviewer 1)
The paper by Dirk Halama et al. is a resubmission of a previous manuscript that carried several major issues. The quality of the current paper, however, is significantly improved. The authors have carefully and systematically improved the manuscript in key areas and addressed the signaled problems.
One major problem that has appeared after the revision is that the reference numbers in the text (citations) do not correspond with the number in the list within the References section - please carefully check all references and renumber them accordingly. The list seems to hold two extra references that are not found within the manuscript.
There are also further minor improvements to be made:
- the authors should introduce a small discussion as to how they decided to use the cutoff value of 50 years for the division into an "old" and a "young" group
- the headings in the abstract should be removed, as per the instructions for authors
- the numbering in the keywords should be removed, as per the instructions for authors
Author Response
Response to Reviewer 1 Comments
The paper by Dirk Halama et al. is a resubmission of a previous manuscript that carried several major issues. The quality of the current paper, however, is significantly improved. The authors have carefully and systematically improved the manuscript in key areas and addressed the signaled problems.
Point 1:
One major problem that has appeared after the revision is that the reference numbers in the text (citations) do not correspond with the number in the list within the References section - please carefully check all references and renumber them accordingly. The list seems to hold two extra references that are not found within the manuscript.
Answer: Thank you very much for your thorough review. We checked the references and corrected the missing link to the reference editor for the section “Statistics”. The small numbers in the discussion are accounted for multiple mentioning.
There are also further minor improvements to be made:
Point 2:
- the authors should introduce a small discussion as to how they decided to use the cutoff value of 50 years for the division into an "old" and a "young" group
Answer: The paragraph “2.5. Physical characteristics” was completed by:
“The cut-off value of 50 years was the result of clustering of vessel values according to age. Data clustering is a method that can form classes of objects with similar characteristics. This method partitions the dataset of vessel diameters into clusters.”
Point 3:
- the headings in the abstract should be removed, as per the instructions for authors
Answer: We apologize, the headings in the abstract are removed.
Point 4:
- the numbering in the keywords should be removed, as per the instructions for authors
Answer: We apologize, the numbers are removed.
Reviewer 2 Report (New Reviewer)
This study by Halama and Colleagues from Leipzig aimed to generate adjusted reference values for angiographic studies. This is an interesting study of technical significance. I have several comments for authors to consider.
1. Given this "retrospective" study analysed vessels from anterior circulation from a tertiary care facility, I suggest authors include "anterior circulation" in the manuscript title, as "Reference values of cerebral artery diameters of anterior circulation by digital subtraction angiography: a tertiary care retrospective study from Germany".
2. For sake of reporting, it is important that the study design, I presume being "retrospective" should be clearly indicated in the title (as above), abstract as well as in methods.
3. One major comment on this study is that this study may not be representative of other populations, as the current study was conducted on a rather homogenous population from Leipzig. There may be inter-ethnic variations which may not be fully appreciated in this study. This should be discussed in the limitations.
4. The Introduction, currently, is rather limited. Suggest expanding the introduction to include some recent studies and references on DSA. Perhaps, the role of DSA in endovascular procedures, specifically in acute ischemic stroke, may be briefly discussed given it is a major procedure routinely performed in neuroangio/INR labs (see Shaban et al 2022).
5. Methods: Please specify the department and hospital - whose data was reviewed. Please clarify how many authors, and years of their experience, examined the angiograms.
6. It is not clear if DSA images were also analysed on other platforms such as Osirix. Please clarify.
7. Please clarify how many subjects were excluded from the study and the specific reasons they were. Have the authors considered a structured reporting template such as STROBE guidelines. Please clarify.
8. I also recommend authors to provide a flowchart indicating the study inclusion or exclusion and various analyses performed. This would improve understanding of the methodological design.
9. The limitations need to be expanded to highlight the methodological limitations of this study.
10. In the conclusion, suggest adding that the conclusions derived from this study should be interpreted in the context of the retrospective study design and study population.
Author Response
Response to Reviewer 2 Comments
This study by Halama and Colleagues from Leipzig aimed to generate adjusted reference values for angiographic studies. This is an interesting study of technical significance. I have several comments for authors to consider.
Point 1:
Given this "retrospective" study analysed vessels from anterior circulation from a tertiary care facility, I suggest authors include "anterior circulation" in the manuscript title, as "Reference values of cerebral artery diameters of anterior circulation by digital subtraction angiography: a tertiary care retrospective study from Germany".
Answer: The title was changed as follows:
“Reference Values of Cerebral Artery Diameters of the Anterior Circulation by Digital Subtraction Angiography: A Retrospective Study.”
The term “tertiary care” is misunderstandable and therefore rarely used in titles. From our point of view, the physical constitution of german patients is not that exceptional that it would need to be addressed in the title (see below).
Point 2: For sake of reporting, it is important that the study design, I presume being "retrospective" should be clearly indicated in the title (as above), abstract as well as in methods.
Answer: The terms „Retrospective/retrospectively“ were added to the title and to the abstract.
Point 3: One major comment on this study is that this study may not be representative of other populations, as the current study was conducted on a rather homogenous population from Leipzig. There may be inter-ethnic variations which may not be fully appreciated in this study. This should be discussed in the limitations.
Answer: The following paragraph was added to “Limitations”:
“In addition, there may be inter-ethnic variations which are not fully appreciated in this study. Leipzig has an increasing percentage of foreigners and immigrants with 10.8% in 2014, 13.4% in 2016 and 16.8 % immigrants in 2021 (State Statistical Office of Saxony) [52]. Most immigrants came from Syria, Ukraine, the Russian Federation, Poland, Iraq, Turkey, Kazakhstan, and Afghanistan. Most of them belong to the European or Asian population. Ethnic minorities carry with them their genetic background and lifestyle habits, which merge with high income countries’ cultures, increasing the risk of developing cardiovascular and metabolic diseases [53]. Our study group compromises a small percentage (<5%) of non-European ethnic-subgroups with neglectable influence on vessel diameters.“
- Stadt Leipzig. Migrantinnen und Migranten, Integration und interkulturelle Aktivitäten in Leipzig. Available online: https://www.leipzig.de/jugend-familie-und-soziales/auslaender-und-migranten/migration-und-integration(accessed on 2022/10/05).
- Tillin, T.; Forouhi, N.; Johnston, D.G.; McKeigue, P.M.; Chaturvedi, N.; Godsland, I.F. Metabolic syndrome and coronary heart disease in South Asians, African-Caribbeans and white Europeans: a UK population-based cross-sectional study. Diabetologia 2005, 48, 649-656, doi:10.1007/s00125-005-1689-3.
Point 4: The Introduction, currently, is rather limited. Suggest expanding the introduction to include some recent studies and references on DSA. Perhaps, the role of DSA in endovascular procedures, specifically in acute ischemic stroke, may be briefly discussed given it is a major procedure routinely performed in neuroangio/INR labs (see Shaban et al 2022).
Answer: The introduction was expanded as follows:
“Digital subtraction angiography (DSA) is considered the gold standard imaging modality in acute cerebrovascular diagnosis. Its role has become increasingly prominent since the endovascular therapy of acute ischemic stroke is a standard of care procedure. It provides diagnostic information regarding intracranial arterial diseases, e.g. atherosclerosis, stenosis, vessel dissection, vasculitis, moyamoya disease and vascular malformations [1]. The requirement of ionizing radiation and the procedural risks of an invasive procedure make it suboptimal for routine vascular assessment. Therefore, data acquisition from healthy subjects in order to create a reference group is challenging.
DSA is superior to computed tomography angiography (CTA) in assessing the severity and the impact of symptomatic cerebral vasospasm (CVS) on subsequent perfusion [2] with indicated endovascular intervention [3]. Mild and moderate vasospasm in distal locations is less detected on CTA. Furthermore, CTA can overestimate the degree of CVS by underestimating the diameter of large cerebral arteries. In previous studies [4,5], the diagnosis was usually made by comparing the DSA on admission with the DSA at the time of suspected CVS. Early angiographic vasospasm on admission was not taken into account [6]. The most commonly used scheme to classify CVS is a reduction in vessel diameters of <25% (mild), 25-50% (moderate), and >50% (severe) [7,8] - or < 30% (grade 1), 30-70% (grade 2) and >70% (grade 3) [9]. Reference values for intracranial arteries are required to standardize CVS classification [10]. …”
- Shaban, S.; Huasen, B.; Haridas, A.; Killingsworth, M.; Worthington, J.; Jabbour, P.; Bhaskar, S.M.M. Digital subtraction angiography in cerebrovascular disease: current practice and perspectives on diagnosis, acute treatment and prognosis. Acta Neurol Belg 2022, 122, 763-780, doi:10.1007/s13760-021-01805-z.
- Yao, Z.; Hu, X.; You, C. Endovascular therapy for vasospasm secondary to subarachnoid hemorrhage: A meta-analysis and systematic review. Clin Neurol Neurosurg 2017, 163, 9-14, doi:10.1016/j.clineuro.2017.09.016.
- Joo, S.P.; Kim, T.S.; Kim, Y.S.; Moon, K.S.; Lee, J.K.; Kim, J.H.; Kim, S.H. Clinical utility of multislice computed tomographic angiography for detection of cerebral vasospasm in acute subarachnoid hemorrhage. Minim Invasive Neurosurg 2006, 49, 286-290, doi:10.1055/s-2006-954826.
- Jabbarli, R.; Pierscianek, D.; Rolz, R.; Darkwah Oppong, M.; Kaier, K.; Shah, M.; Taschner, C.; Monninghoff, C.; Urbach, H.; Beck, J.; et al. Endovascular treatment of cerebral vasospasm after subarachnoid hemorrhage: More is more. Neurology 2019, 93, e458-e466, doi:10.1212/WNL.0000000000007862.
- Tjerkstra, M.A.; Verbaan, D.; Coert, B.A.; Post, R.; van den Berg, R.; Coutinho, J.M.; Horn, J.; Vandertop, W.P. Large practice variations in diagnosis and treatment of delayed cerebral ischemia after subarachnoid hemorrhage. World Neurosurg 2022, doi:10.1016/j.wneu.2022.01.033.
- Jabbarli, R.; Reinhard, M.; Shah, M.; Roelz, R.; Niesen, W.D.; Kaier, K.; Taschner, C.; Weyerbrock, A.; Van Velthoven, V. Early Vasospasm after Aneurysmal Subarachnoid Hemorrhage Predicts the Occurrence and Severity of Symptomatic Vasospasm and Delayed Cerebral Ischemia. Cerebrovasc Dis 2016, 41, 265-272, doi:10.1159/000443744.
- Janjua, N.; Mayer, S.A. Cerebral vasospasm after subarachnoid hemorrhage. Curr Opin Crit Care 2003, 9, 113-119, doi:10.1097/00075198-200304000-00006.
- Samagh, N.; Bhagat, H.; Jangra, K. Monitoring cerebral vasospasm: How much can we rely on transcranial Doppler. J Anaesthesiol Clin Pharmacol 2019, 35, 12-18, doi:10.4103/joacp.JOACP_192_17.
- Kerz, T.; Boor, S.; Beyer, C.; Welschehold, S.; Schuessler, A.; Oertel, J. Effect of intraarterial papaverine or nimodipine on vessel diameter in patients with cerebral vasospasm after subarachnoid hemorrhage. Br J Neurosurg 2012, 26, 517-524, doi:10.3109/02688697.2011.650737.
- Merkel, H.; Lindner, D.; Gaber, K.; Ziganshyna, S.; Jentzsch, J.; Mucha, S.; Gerhards, T.; Sari, S.; Stock, A.; Vothel, F.; et al. Standardized Classification of Cerebral Vasospasm after Subarachnoid Hemorrhage by Digital Subtraction Angiography. J Clin Med 2022, 11, doi:10.3390/jcm11072011.
Point 5: Methods: Please specify the department and hospital - whose data was reviewed. Please clarify how many authors, and years of their experience, examined the angiograms.
Answer: The “Department of Neuroradiology at the University of Leipzig Medical Center” was added to methods.
The angiograms were analyzed by one consulting physician in neuroanatomy and radiology with three years of angiographic experience (lines 132 – 134). Ambiguous angiograms were discussed and double-checked by an interventional neuroradiologist with more than three years of experience).
Point 6: It is not clear if DSA images were also analysed on other platforms such as Osirix. Please clarify.
Answer: Lines 162-163:
“All values were only measured on a diagnostic workstation (syngo.plaza, VB30C, Siemens Health Care, Erlangen, Germany) according to a defined protocol.”
No other platform was used.
Point 7: Please clarify how many subjects were excluded from the study and the specific reasons they were. Have the authors considered a structured reporting template such as STROBE guidelines. Please clarify.
Answer: Supplemented to “2.1. Study Design and Recruitment” (lines 84 - 85):
“This retrospective study was performed according to the “Strengthening the Reporting of Observational Studies in Epidemiology (STROBE)” guidelines [12].”
The numbers of the excluded digital subtraction angiographies and the exclusion criteria are presented in detail in the flow chart (Supplement 1, Point 8).
Point 8: I also recommend authors to provide a flowchart indicating the study inclusion or exclusion and various analyses performed. This would improve understanding of the methodological design.
Answer: Please, find attached a detailed flow chart as Supplement 1.
Supplement 1: Flow Chart of study inclusion and exclusion criteria.
DSA digital subtraction angiography, SAH subarachnoid hemorrhage.
Point 9: The limitations need to be expanded to highlight the methodological limitations of this study.
Answer: The limitations were expanded as follows (lines 477 – 497):
“This retrospective study with a focus on imaging findings has several limitations. First of all, a retrospective analysis with key word research in Radiology Information Systems is imprecise. Novel scheduling tools make retrospective data collection easier in future studies. The requirement of ionizing radiation and the procedural risks of an invasive procedure make it challenging to analyze angiograms of healthy humans in order to gain reference values. Most patients suffered from intracranial aneurysms with or without previous SAH. Only a small fraction of 15 patients did not suffer from any vascular disease.
In addition, there may be inter-ethnic variations …”
Point 10: In the conclusion, suggest adding that the conclusions derived from this study should be interpreted in the context of the retrospective study design and study population.
Answer: The conclusion was complemented as follows:
The present study provides gender-, side- and age-rectified reference values of intracranial arteries based on nine standardized measuring sites for a European single-center study population. The presented reference values and nomograms provide a basis for a threshold-based classification of CVS with the need for further evaluation.

Round 2
Reviewer 2 Report (New Reviewer)
The authors have done a great job in addressing the concerns raised previously. I recommend the manuscript be accepted. No further comments.
This manuscript is a resubmission of an earlier submission. The following is a list of the peer review reports and author responses from that submission.
Round 1
Reviewer 1 Report
The authors have submitted a very interesting manuscript on the reference values for intracranial vessels (anterior circulation). The paper is rich in data and has a relatively large sample size. However, there are multiple flaws in the manuscript that should be corrected before considering publication:
- the patient selection criteria are unclear and difficult to read (lines 61-72) - please provide either a flow chart of patient selection and exclusion or a clear set of inclusion/exclusion criteria; also, some information from the first paragraph is repeated in the numbered list of exclusion criteria - please address this issue.
- Table 2 is difficult to interpret and should be reworked so as to make sense to the reader. Also, the "group of age" should be defined.
- The topic of the paper is reference values of cerebral arteries diameters and the authors repeatedly mention that vessels (segments) with aneurysms are excluded from the study lot. However, a significant part of the manuscript is then dedicated to the location of said aneurysms (sections 3.3, 3.4, a good part of the Discussions and some Supplementary materials). Since the title and the declared scope of the paper point to vessel diameter and normal values, these sections should be eliminated from the manuscript as they are outside the objective of the paper. Otherwise, the manuscript should be rewritten with the appropriate study design.
- The grey area is not showing in Figure 2 - please correct it.
- Figure 3 is not introduced in the text and is not properly explained in terms of the meaning of the colours and dotted/straight lines.
- "Women have a higher lifetime risk for stroke [...], although stroke is more common among men. " - This is a misquote of the source and it is phrased in a questionable manner - please reformulate.
- Paragraphs 3 and 4 of Discussion have no point to be made. Please justify their presence by coming to a point at the end of the paragraph, otherwise they appear to be just random facts.
- The confusing style of the authors is resurfacing again in the 5th paragraph of discussions where they start off by saying that CBF, hemodynamics, and morphology vary by age and other parameters and end up relating that there is no correlation of the diameter with age and some vessels don't ever change. What does this mean? Is it a controversy in the literature? Is this the authors' point of view? Is is settled or is there still some debate in this? How does the present study contribute to resolving this issue?
- line 375 - the authors start off by talking about congenital variations and then they refer to aneurysm reports - is the aneurysm considered a congenital variation? What about the configuration of the CoW? The interpretation of the cited studies is ambiguous in this paragraph and needs to be redone.
- "Aneurysms are more commonly located in the anterior than in the posterior circulation, with only 10-15%" - what does this mean? 10-15% of aneurysms are in the posterior circulation? or the anterior one? or is this the difference between anterior and posterior?
- it is unclear how the authors have selected the studies in Table 4. There are plenty other relevant studies that describe the diameter of each segment of the mentioned vessels; also, the present study is not included in the table for comparison. What is the purpose of the table? It does not seem to refer to segment diameters and includes various types of studies that will obviously carry relatively different results.
- what is the deficient standardized CVS scheme that is mentioned in line 418? please add a reference. if there is none, the authors should mention it is lacking or inexistent and not deficient/incomplete/insuficient etc...
- the authors mentioned that the height of the males in their study was superior to that of the women, therefore accounting for the differences in gender. was the weight also considered? please provide the data for comparison of males vs females lots depending on height and weight of the subjects and use statistical tests to demonstrate that the differences between M and F are actually caused by differences in height/weight.
- what are the falsifying influencing factors that were excluded for the nomograms? (line 56) Later on, just aneurysms are mentioned.
- how was the cutoff value of 50 years old been determined?
Overall, the manuscript can benefit from a thorough rewrite to reshape it into a more robust version in terms of methodology, presentation of data, and discussions. Thank you for the opportunity for reviewing this paper and best regards to the authors for their efforts and work put into the manuscript.
Reviewer 2 Report
This study provides reference values regarding the diameters of intracranial arteries. The authors have to be congratulated on a major, detailed and very carefully conducted study.
The manuscript is of value as a reference and despite the authors arguing its use to guide classification of vasospasm and devise implantation, this would be in general terms. However, due to variations and the range of values and in addition the influence of other disease factors such as atherosclerosis, then practically applications have to be individually tailored.